# Development and Optimization of Broadband Acoustic Metamaterial Absorber Based on Parallel–Connection Square Helmholtz Resonators

**DOI:** 10.3390/ma15103417

**Published:** 2022-05-10

**Authors:** Enshuai Wang, Fei Yang, Xinmin Shen, Haiqin Duan, Xiaonan Zhang, Qin Yin, Wenqiang Peng, Xiaocui Yang, Liu Yang

**Affiliations:** 1College of Field Engineering, Army Engineering University of PLA, Nanjing 210007, China; wangenshuai0823@126.com (E.W.); 19962061916@163.com (F.Y.); dhq1135168523@163.com (H.D.); zxn8206@163.com (X.Z.); dafengyinqin@126.com (Q.Y.); sueecces@126.com (L.Y.); 2College of Aerospace Science and Engineering, National University of Defense Technology, Changsha 410073, China; 3Engineering Training Center, Nanjing Vocational University of Industry Technology, Nanjing 210023, China; 2019101052@niit.edu.cn; 4MIIT Key Laboratory of Multifunctional Lightweight Materials and Structures (MLMS), Nanjing University of Aeronautics and Astronautics, Nanjing 210016, China

**Keywords:** acoustic metamaterial absorber, parallel–connection square Helmholtz resonators, low frequency noise control, particle swarm optimization algorithm, finite element simulation model, adjustable frequency spectrum, standing wave tube verification

## Abstract

An acoustic metamaterial absorber of parallel–connection square Helmholtz resonators is proposed in this study, and its sound absorption coefficients are optimized to reduce the noise for the given conditions in the factory. A two–dimensional equivalent simulation model is built to obtain the initial value of parameters and a three–dimensional finite element model is constructed to simulate the sound absorption performance of the metamaterial cell, which aims to improve the research efficiency. The optimal parameters of metamaterial cells are obtained through the particle swarm optimization algorithm, and its effectiveness and accuracy are validated through preparing the experimental sample using 3D printing and measuring the sound absorption coefficient by the standing wave tube detection. The consistency between the experimental data and simulation data verifies feasibility of the proposed optimization method and usefulness of the developed acoustic metamaterial absorber, and the desired sound absorption performances for given conditions are achieved. The experimental results prove that parallel–connection square Helmholtz resonators can achieve an adjustable frequency spectrum for the low frequency noise control by parameter optimization, which is propitious to promote its application in reducing the noise in the factory.

## 1. Introduction

Noise in the large factories not only harm the health of operators and workers [1], but also increase the error rate during the operation process [2], both of which would inevitably result in the decrease of production efficiency and the increase of manufacturing cost [3]. Owing to the inevitable noise resulted from the working equipment, an effective method to reduce the damage of noise is to place the sound absorbing materials around the noise source. Thus, many kinds of sound absorbing materials and structures have been developed, such as the porous materials [4], microperforated panel absorber [5], acoustic metamaterial [6], et al. These developed sound absorbing materials and structures can be helpful for noise suppression.

Among the presently developed sound absorbing materials and structures, the acoustic metamaterial is the most promising candidate to be widely applied in reducing the noise generated from working equipment in the factory, because the noise is mainly in the low–frequency range [7,8,9,10]. Hedayati and Lakshmanan [7] had proposed the pneumatically–actuated acoustic metamaterials based on Helmholtz resonator, which could achieve the acoustic bandgap shifted from a frequency band of 150–350 Hz to that of 300–600 Hz. An acoustic slow–wave effect metamaterial muffler for noise control of the HVDC (high voltage direct current) converter station, developed by Yang et al. [8], could obtain a broadband quasi–perfect absorption of noise from 600 to 900 Hz. Zhang et al. [9] developed the light–weight large–scale tunable metamaterial panel for the low–frequency sound insulation, and the multiple local resonance caused sound transmission loss improvements over the traditional mass law.

The sound absorption performance of acoustic material is mainly determined by its structural parameters, which indicates that optimization of its parameter is an important step to promote its actual application for the variable conditions [11,12,13,14,15]. Gao et al. [11] had conducted the optimal design of broadband quasi–perfect sound absorption of composite hybrid porous metamaterial by using the TLBO (teaching–learning–based optimization) algorithm. The systematic design and realization of double–negative acoustic metamaterials through topology optimization was carried out by Dong et al. [12], and the inverse design of acoustic metamaterials based on the machine learning by using a Gauss–Bayesian model was achieved by Zheng et al. [13]. Chen et al. [14] had optimized the acoustic metamaterial cloaks under uncertainty and developed the scalable approximation and optimization methods to solve this problem. These research achievements prove that optimization can improve the sound absorption effect.

A Helmholtz resonator can absorb low frequency sound through a thermal viscosity effect and a thermal conductivity effect, and the parallel square Helmholtz resonator has been proved as a feasible and practical acoustic absorber because it has the advantages of high sound absorption efficiency, large sound absorption width in low frequency range, simple structure, high extensibility, and so on, which means it has already been applied in some noise reduction fields [16,17,18]. Each square Helmholtz resonator can obtain an absorption peak with suitable parameters, and the parallel square Helmholtz resonators can achieve a wide sound absorption range with high sound absorption efficient by superposition of these absorption peaks [19]. In order to effectively obtain the desired sound absorption frequency range with the required sound absorption performance, the structural parameters of parallel–connection square Helmholtz resonators should be designed and optimized reasonably [20].

However, there are three difficulties in development and optimization of parallel square Helmholtz resonators. First of all, the sound absorption coefficient of parallel square Helmholtz resonators is difficult to calculate accurately, because the theoretical sound absorption model is imprecise for some simplifications, approximations, hypotheses, and abbreviations. Although the finite element simulation method can achieve high prediction accuracy, it takes a long time. Secondly, many structural parameters need to be optimized for the parallel square Helmholtz resonators, and an increase in the number of parameters not only raises the difficulty in design and optimization but also extends the calculation or simulation time. Thirdly, an effective optimization process requires not only a suitable utilized algorithm but also to be determined by the selected initial value, the given constraint conditions, and the desired optimization objectives.

Therefore, the parallel–connection square Helmholtz resonator is investigated and optimized in this research, including the structural design of the metamaterial cell, the construction of finite element model, the optimization of structural parameters, the fabrication and detection of optimal sample, and the analysis of sound absorption performance. Firstly, a 4–groups parallel metamaterial cell with 4 square Helmholtz resonators in each group is given as a research target, which is used to explain the meaning of studied parallel–connection square Helmholtz resonators. Secondly, the finite element simulation model of the metamaterial cell is constructed, which is utilized to calculate the sound absorption coefficient and analyze the sound absorption procedures. Thirdly, for the 3 requirements of noise reduction with the given conditions, the structural parameters of the metamaterial cell are optimized respectively using the swarm optimization algorithm. Fourthly, the corresponding experimental samples are fabricated through 3D printing and further detected by standing wave tube measurement. Finally, the sound absorption performance of parallel–connection square Helmholtz resonators is analyzed.

## 2. Structural Design of the Metamaterial Cell

The metamaterial cell of parallel–connection square Helmholtz resonators is shown in Figure 1, which takes the 4–groups parallel metamaterial cell with 4 square Helmholtz resonators in each group as a research target. Different from the common parallel square Helmholtz resonators, the Helmholtz resonators in this metamaterial cell are divided into 4 groups, and diameters of the hole for the Helmholtz resonators in one group are equal. As shown in the Figure 1b, the resonators are labeled as R1 to R16. For each single resonator, there are 4 parameters, including diameter of the hole d, length of the aperture l, side length of the cavity s, thickness of the cavity D, as shown in the Figure 1c. Meanwhile, t is thickness of the wall. In this study, t is set as 2 mm, which takes the fabrication requirement in the 3D printing process into consideration.

The common acoustic metamaterial consisting of n parallel square Helmholtz resonators has 4n structural parameters, which indicates that there will be 64 structural parameters as the number of Helmholtz resonators is 16. Optimization efficiency will decrease exponentially with an increase in the number of parameters, no matter whether it is by the theoretical modeling or the finite element simulation. Therefore, it is essential to give some limits for the parameters.

Firstly, the resonators can be divided into four groups, as shown in Figure 1b. For the four resonators in one group, the diameter of the hole for each resonator is equal. The perforation rate r is defined as the ratio of total area of the holes with the same diameter to the whole area of the metamaterial cell, as shown in Equation (1), which is similar to the definition of perforation rate for the microperforated panel absorber. It had been proved that the perforation rate should be in the range of 0.5~0.9% to obtain efficient sound absorption for the Helmholtz resonator [18], and it can be calculated by the Equation (1) for the investigated Helmholtz resonator in Figure 1. Definitions of the symbols in the Equation (1) are the same as those in the Figure 1. If the perforation rate is larger than 0.9%, the resonance frequency will exceed the low–middle frequency range, which cannot show the advantages of the Helmholtz resonator in the low frequency region. Meanwhile, when the perforation rate is smaller than 0.5%, the thermal viscous effect and thermal conduction effect will be weakened, which leads to the decrease of the peak sound absorption coefficient. Thus, the perforation rate should be in reasonable range.
(1)$$\begin{array}{cc} r_{i}=\frac{4\cdot \pi {\left(d_{i}/2\right)}^{2}}{{\left(4s_{i}+5t\right)}^{2}} & i=1,2,3,4 \end{array}$$

Secondly, the side length of the cavity for each resonator is uniform. Meanwhile, the size of the metamaterial cell should be no more than 70.7 mm ($100/2\cdot \sqrt{2}$ mm), because the requirement of size of the sample in the standing wave tube measurement is limited to ϕ100 mm. Therefore, the size of the metamaterial cell is set to 70 mm in this research. Moreover, side length of the cavity s can be derived as 15 mm ($\left(70-2\times 5\right)/4$ mm). Furthermore, diameters of the hole for 4 groups of resonators in the metamaterial cell can be calculated as 3.54, 3.87, 4.18, and 4.47 mm as the selected perforation rates are 0.5%, 0.6%, 0.7%, and 0.8%, respectively.

Thirdly, the thickness of the cavity for each resonator is equal; otherwise, there will be space waste for the investigated metamaterial cell. In most conditions, the available space to install the sound absorber is limited. It had been proved that the limit of space is the most important factor to decide the range of effective sound absorption frequency and distribution of sound absorption coefficient [21]. Normally speaking, it would achieve the better sound absorption performance with larger available space. Therefore, the uniform thickness of the metamaterial cell in Figure 1 is set to be equal to the available space in the given condition.

Therefore, parameters of the metamaterial cell of the parallel–connection square Helmholtz resonators are summarized in Table 1. It can be found that, except thickness of the cavity D (which is equal to the available space subtracts the double wall), the parameters that need to be optimized are the lengths of the aperture $l_{i}\left(i=1,2,\ldots ,16\right)$. By this method, the basic metamaterial cell of the proposed parallel–connection square Helmholtz resonators is constructed, and the structural parameters that need to be optimized are established, which provides a foundation for further investigation and study. Through a decrease in the number of parameters to be optimized, optimization difficulty is reduced, which is propitious for improving research efficiency.

## 3. Construction of the Finite Element Model

### 3.1. Three–Dimensional Finite Element Model

The built three–dimensional finite element model for the parallel–connection square Helmholtz resonators is shown in Figure 2. In the geometric model, as shown in Figure 2a, the perfect matching layer is used to simulate the air domain with the total absorption of the incoming sound wave, and the background acoustic field is utilized to simulate the acoustic source with the defined type and propagation direction. The other part in Figure 2a is a model of air domain of parallel–connection square Helmholtz resonators, which corresponds to the wall structure in Figure 1a. Afterwards, the finite element mesh model is constructed through gridding with free tetrahedral mesh, as shown in Figure 2b. Through calculating the sound pressure distribution in the metamaterial cell, as shown in Figure 2c, the sound absorption coefficient of the detected acoustic absorber can be derived within a certain frequency range. In the finite element model, when the frequency of incident sound wave is consistent with the resonance frequency of resonant cavity, there will be frequent expansion and compression for air in the cavity, which can result in the reciprocating motion with high speed for the air column in the aperture. The thermal viscosity effect between air column in the aperture and the boundary and the thermal conductivity effect in the wall can lead to losses of acoustic energy, which achieves the sound absorption effect.

The parameters used in the three–dimensional finite element model are as follows: size of the largest unit, 2 mm; size of the smallest unit, 0.02 mm; maximum growth rate of neighboring unit, 1.3; curvature factor, 0.2; resolution ratio of the narrow area, 1; mesh type, free tetrahedron mesh; layer number of the boundary area, 8; stretching factor of the boundary layer, 1.2; regulatory factor of the thickness of the boundary layer, 1.

### 3.2. Two–Dimensional Equivalent Simulation Model

Although the three–dimensional model can achieve the sound absorption coefficient based on the finite element simulation, its usual calculation time is over 10 h, sometimes even more than 100 h. In order to improve research efficiency, a two–dimensional equivalent simulation model is built, which is utilized to establish the initial value for each resonator of the parallel–connection square Helmholtz resonator, as shown in the Figure 3. The two–dimensional model is a rotational symmetry structure, and the square resonator is equivalent to the cylindrical cavity with the same sectional area. The side length of the cavity s is 15 mm in this study, so the equivalent diameter of the cylindrical cavity $d^{\prime }$ is 16.93 mm ($\sqrt{4/\pi }\cdot 15\approx 16.93$ mm). Similar to the geometric model in Figure 2a, the two–dimensional geometric model in Figure 3a includes the perfect matching layer and background acoustic field, and the other section is model of the air domain of the single resonators, which corresponds to the wall structure in Figure 1c. Afterwards, the finite element mesh model is constructed through gridding with free triangle mesh, as shown in Figure 3b. For the perfect matching layer, the mesh is obtained through mapping. Meanwhile, in order to improve the simulation accuracy, boundary areas are further specified, which can be observed in Figure 3b. Later, through setting the interesting frequency range and other parameters, the acoustic field of the whole model can be obtained, as shown in Figure 3c, which can be utilized to calculate the sound absorption coefficient in simulation.

The finite element mesh model is constructed through gridding with the free triangle mesh, as shown in Figure 3b, and the boundary layers are further specified. Through the calculation of sound pressure distribution in the two–dimensional equivalent simulation model, as shown in Figure 3c, the sound absorption coefficient of the investigated single resonator can be obtained. Taking a series of structural parameters, for example, the distributions of its theoretical sound absorption coefficients along with an increase in length of the aperture l are exhibited in Figure 4, where the parameters are D=40 mm, d=3.54 mm, $d^{\prime }=16.93$ mm, T=25 mm and t=2 mm. It can be found that the peak absorption frequency shifts to the low–frequency direction and the corresponding maximum absorption coefficient decreases along with the increase of l when the other parameters are established. Therefore, through controlling the value of the l, the peak absorption frequency and the maximum absorption coefficient can be adjusted as needed, which can be used to investigate the sound absorption coefficient of each resonator and select a suitable initial value for the metamaterial cell of parallel–connection square Helmholtz resonators. The reasonable initial value can obviously improve optimization efficiency.

## 4. Optimization of the Structural Parameters

### 4.1. Optimization Objectives

The three requirements of noise reduction in the given factories are taken as the optimization objectives in this research, which include the size limitation of acoustic absorber, the interested frequency range, and the required sound absorption performance. The optimization objectives are summarized and shown in Table 2, and they are labeled as condition–1, condition–2, and condition–3, successively. Taken the condition–1, for example, the size of the acoustic absorber should be no more than 30 mm, and the sound absorption coefficients for each frequency point in the concerned frequency range of 700–1000 Hz should be larger than 0.8. Besides this requirement, it is better if the average sound absorption coefficient in the concerned frequency range is larger. Thus, the requirement of sound absorption coefficients at each frequency point larger than 0.80 can be treated as the constraint condition, and the corresponding average sound absorption coefficient can be considered as the optimization goal. As mentioned above, along with an increase in the requirement of sound absorption performance and shift of the concerned frequency range to the low frequency direction, the needed space to install the acoustic absorber raises gradually, which is consistent with the optimization objectives in Table 2.

### 4.2. Intelligent Optimization Algorithm

Many intelligent optimization algorithms had been chosen in optimizing the structural parameters of acoustic metamaterial, such as the level set–based topology optimization method utilized by Noguchi et al. [22], genetic algorithm used by Li et al. [23], reinforcement learning applied by Shah et al. [24], and topology optimization adopted by Dong et al. [12,25]. In this study, the particle swarm optimization algorithm [26] with the given initial values is used to optimize structural parameters of the investigated metamaterial cell. Although the initial values of parameters to be optimized are randomly generated in most optimization algorithms, they are screened firstly by the two–dimensional equivalent simulation model in Figure 3, which aims to improve research efficiency. The random generated initial values may be far away from the optimal solutions, which indicates that the optimum iterative procedure needs many loops and will take a long time to reach the optimal solution.

The flow chart of the optimization program is shown in Figure 5, and its key procedures are calculation of the initial value for parameters by the two–dimensional equivalent simulation model and derivation of sound absorption coefficients of the metamaterial cell through a three–dimensional finite element model. The parameters are iteratively updated through a particle swarm optimization algorithm. According to the optimization objectives of size limitation of the acoustic absorber L and the interested frequency range $\left[f_{\min},f_{\max}\right]$, the initial values for length of the aperture $l_{i}$ can be achieved. Through comparing the achieved sound absorption coefficients with the required sound absorption performance and contrasting the satisfactory results with the present preserved optimal results, the optimal results are continuously improved. In order to avoid excessive iteration times taking too much time, the additional iteration times are set to 20 when the first satisfactory results are obtained. Therefore, the optimal results obtained in this research are the optimal solution with some constraints instead of the final global optimum (which may require a very long time to obtain), which takes both the desired requirement and the optimization efficiency into consideration.

### 4.3. Initial Values of Parameters

According to the optimization process in Figure 5, initial values of parameters for the proposed metamaterial cell are achieved on the basis of the desired optimization objectives in Table 2 and the two–dimensional equivalent simulation model in Figure 3, as shown in Table 3. Taking the condition–1, for example, the concerned frequency range is 700–1000 Hz, so the desired peak absorption frequencies for the 16 single resonators in the metamaterial cell are 700, 720, 740, 760, 780, 800, 820, 840, 860, 880, 900, 920, 940, 960, 980, and 1000 Hz, successively. Thickness of the cavity D is 24 mm (*L* − 2*t* = 30 − 2 × 2). According to the structural parameters of the single resonator in the Figure 3a, the initial values for length of the aperture $l_{i}$ can be achieved. Distributions of sound absorption coefficients of each single resonator for the three conditions are shown in the Figure 6. It can be observed that length of the aperture l decreases gradually along with increase of the desired absorption peak frequency when the diameter of the hole d and the thickness of the cavity D are given. Taking the condition of d=3.54 mm and D=26 mm in the condition–1, for example, the lengths of the aperture l are 7.57, 6.95, 6.42, and 5.94 mm corresponding to the desired absorption peak frequencies of 700, 720, 740, and 760 Hz. Meanwhile, it can be calculated that the average lengths of the aperture are 5.99, 5.04, and 5.26 mm for the 3 conditions, respectively. The exhibited characters are consistent with the common principle that both the long aperture and the thick cavity are conducive to achieving a peak absorption in the low frequency region. When the desired absorption peak frequency is 700 Hz, the parameters for the 3 conditions are l=7.57 mm, d=3.54 mm, D=26 mm; l=5.62 mm, d=3.87 mm, D=36 mm; and l=4.37 mm, d=4.18 mm, D=46 mm, respectively. When the available space to install the acoustic absorber is small, the diameter of the hole d has to be small and the length of the aperture l must be large to achieve the certain absorption peak frequency. Achievement of initial values can be propitious for improving research efficiency.

### 4.4. Optimization Results

The obtained optimal parameters for the 3 conditions are summarized in Table 4, and the corresponding sound absorption performances obtained by the three–dimensional finite element model are exhibited in Figure 7. Meanwhile, the corresponding absorption peak frequencies for each single resonator are calculated and summarized in Table 4. It can be found that the optimal parameters in Table 4 are obviously different from the initial values of the parameters in Table 3 because the sound absorption effect of the metamaterial cell is obtained through the coupling action of all the resonators instead of simple superposition of each single resonator. However, it can also be observed that some of the optimal parameters are close to their initial values, which indicates that the optimization procedure is simplified, and the optimization efficiency is improved. Secondly, it can be found that some resonators have the same or very close absorption peak frequency, such as 925 Hz for $l_{12}=5.01$ mm and 926 Hz for $l_{13}=6.12$ mm in condition–1, 830 Hz for $l_{12}=3.85$ mm and 830 Hz for $l_{13}=4.74$ mm in the condition–2, and 719 Hz for $l_{12}=3.94$ mm and 720 Hz for $l_{13}=4.82$ mm in the condition–3. It is interesting to note that these overlapping absorption peak frequencies appear at the last resonator of the front group and the first resonator of the next group, which further prove that there is coupling action among the varied resonators. Thirdly, it can be found that the absorption peak frequency of the boundary resonator exceeds the desired frequency range, such as 1020 Hz and 1050 Hz in the condition–1; 574, 589, 918, and 957 Hz in the condition–2; and 485, 495, 814, and 829 Hz in the condition–3. Therefore, the selected absorption peak frequency should enlarge to realize the desired sound absorption frequency range. There is also interaction among the resonators. The interaction is stronger when the parameters of the resonators are close to each other, and it is weaker when the parameters of the resonators are far away from each other.

Moreover, distributions of the sound pressure at each resonance frequency for the three optimized metamaterial cells are exhibited in Figure 8. It can be observed that each resonator corresponds to a resonance frequency, and final sound absorption performance of the proposed metamaterial cell of 4–groups parallel square Helmholtz resonators is achieved through the combined effects of the 16 single resonators. Meanwhile, it can be found that for each resonance frequency, the sound absorption effect of the resonant cavity is not always the best compared to those of the other cavities. Taking *f* = 714 Hz in Figure 8a, for example, the resonant cavity is R5 referring to Figure 1b, but its sound absorption effect is worse than those of the cavities of R2, R3, R4, and R6, which can be judged from distribution of the sound pressure for this resonance frequency in Figure 8a. The major reason for this phenomenon is that the obtained optimal length of the aperture $l_{5}$ in Table 4 is 9.11 mm, which is far away from that of the prospective value 7.01 mm. It can also be judged from Figure 4 that the sound absorption effect is affected by the varied resonance frequency.

## 5. Fabrication and Detection of the Optimal Sample

### 5.1. Preparation of Experimental Sample

According to the obtained optimal parameters for the three investigated conditions in the Table 4, the corresponding three experimental samples are prepared through the Form3 low force stereolithography 3D printer (Formlabs Inc., Boston, MA, USA) based on light curing technology. The utilized 3D printer and the fabricated experimental sample are shown in Figure 9. A three–dimensional structural model of the optimized metamaterial cell is constructed in the Solidworks 3D modeling software (Dassault Systèmes SOLIDWORKS Corp., Waltham, MA, USA) according to the optimal parameters in Table 4, and the model is further handled by the preprocessing software to achieve the available model for the 3D printer. After 3D printing, the prepared samples are further cleaned and consolidated, and the final experimental samples for the three investigated conditions are shown in Figure 9b–d, respectively. The diameter of the sample in Figure 9 is Φ100 mm, which meets the requirement of the following standing wave tube detector to measure the sound absorption coefficient within the low frequency range of 200–1600 Hz. It can be observed that each single resonator consists of an aperture and the rear cavity, which can realize the noise reduction effect on the basement of their corresponding acoustic impedances. The residues on the surfaces shown in Figure 9 are resulted during removal of the supports generated in the 3D printing process.

### 5.2. Detection of Sound Absorption Coefficient

The prepared experimental samples are detected by AWA6290T standing wave tube detector (Hangzhou Aihua instruments Co., Ltd., Hangzhou, Zhejiang, China), and distributions of actual sound absorption coefficients for the three investigated conditions are obtained and exhibited in Figure 10. The dark horizontal dotted lines in Figure 10 are α=0.8 and α=0.85, respectively. Meanwhile, the green vertical dotted lines, blue vertical dotted lines, and red vertical dotted lines represent the desired frequency ranges of [500 Hz, 800 Hz], [600 Hz, 900 Hz], and [700 Hz, 1000 Hz] for the three investigated conditions, respectively. It can be found that actual sound absorption coefficients of the three investigated metamaterial cells can satisfy the requirements of noise reduction for the three given conditions. Meanwhile, comparing the corresponding distributions of sound absorption coefficients of the metamaterial cells obtained by three–dimensional finite element simulation in the Figure 7, it can be observed that the simulation data are consistent with the corresponding experimental data, which proves the effectiveness and accuracy of this proposed optimization method. In addition, each frequency point in the frequency range of interest can meet the requirement in Table 3, and the average sound absorption coefficients of the three investigated metamaterial cells are 0.9271 in the [700 Hz, 1000 Hz], 0.9157 in the [600 Hz, 900 Hz], and 0.9259 in the [500 Hz, 800 Hz], respectively, all of which can achieve a broadband sound absorption performance and a large sound absorption coefficient in the low frequency range. These are coupling effects of each resonator in the metamaterial cell.

## 6. Analysis and Discussion

A theoretical sound absorption coefficient of the studied parallel–connection square Helmholtz resonators can be obtained according to the Electro–Acoustic Theory [4,27,28,29,30,31], as shown in the Equation (2). Here α is the theoretical sound absorption coefficient; Z is the total acoustic impedance of the investigated metamaterial cell; $\rho_{0}$ is the density of the air; $c_{0}$ is the sound velocity in the air.
(2)$$\alpha =1-\left|\frac{Z/\rho_{0}/c_{0}-1}{Z/\rho_{0}/c_{0}+1}\right|$$

The total acoustic impedance of the metamaterial cell Z can be obtained by parallel connection of the single resonators, as shown in Equation (3). Here $Z_{n}$ is the acoustic impedance of the nth single resonator, which includes the acoustic impedance of the aperture $Z_{nm}$ and the acoustic impedance of the rear cavity $Z_{nc}$, as shown in Equation (4).
(3)$$Z=1/\sum_{n=1}^{16}\left(1/Z_{n}\right)$$
(4)$$Z_{n}=Z_{nm}+Z_{nc}$$

The acoustic impedance of the aperture $Z_{nm}$ can be derived by Equation (5) based on the Euler equation [28]. Here ω is the sound angular frequency; $l_{n}$ is the length of the aperture; $\sigma_{n}$ is the perforation ratio; $B_{1}\left(\eta_{n}\sqrt{-i}\right)$ and $B_{0}\left(\eta_{n}\sqrt{-i}\right)$ are the first order and zero order Bessel functions of the first kind, respectively; $\eta_{n}$ is the perforation constant, which can be obtained by the Equation (6); μ is the dynamic viscosity coefficient of the air; $d_{n}$ is the diameter of the hole.
(5)$$Z_{nm}=i\frac{\omega \rho_{0}l_{n}}{\sigma_{n}}{\left[1-\frac{2B_{1}\left(\eta_{n}\sqrt{-i}\right)}{\left(\eta_{n}\sqrt{-i}\right)\cdot B_{0}\left(\eta_{n}\sqrt{-i}\right)}\right]}^{-1}+\frac{\sqrt{2}\mu \eta_{n}}{\sigma_{n}\cdot d_{n}}+i\frac{0.85\omega \rho_{0}\cdot d_{n}}{\sigma_{n}}$$
(6)$$\eta_{n}=d_{n}\sqrt{\rho_{0}\omega /4/\mu }$$

The acoustic impedance of rear cavity $Z_{nc}$ can be achieved through the impedance transfer formula [29], as shown in the Equation (7). Here $Z_{nce}$ is the effective characteristic impedance of the air in the cavity, which can be obtained by the Equation (8); $k_{nce}$ is the effective transfer constant of the air in the cavity, which can be achieved by the Equation (9); D is the thickness of the cavity.
(7)$$Z_{nc}=-iZ_{nce}\cot\left(k_{nce}D\right)$$
(8)$$Z_{nce}=\sqrt{\rho_{0e}/C_{0e}}$$
(9)$$k_{nce}=\omega \sqrt{\rho_{0e}C_{0e}}$$

In the Equations (8) and (9), $\rho_{0e}$ and $C_{0e}$ are the effective density and effective volumetric compressibility of air, respectively, which can be obtained by the Equations (10) and (11). Here v can be calculated by the Equation (12); a and h are the side lengths of the cavity section, which is equal to s in the Figure 1c; $\alpha_{x}^{}=\left(x+1/2\right)\pi /a$ and $\beta_{y}^{}=\left(y+1/2\right)\pi /h$ are the intermediate calculation coefficients; $P_{0}$ is the standard atmospheric pressure under normal temperature; γ is the specific heat rate of the air; $v^{\prime }$ can be obtained through the Equation (13), and κ and $C_{v}$ are thermal conductivity and specific heat capacity at the condition of constant volume, respectively.
(10)$$\rho_{0e}=\rho_{0}\frac{va_{}^{2}h_{}^{2}}{4i\omega }{\left\{\sum_{x=0}^{\infty }\sum_{y=0}^{\infty }{\left[\alpha_{x}^{2}\beta_{y}^{2}\left(\alpha_{x}^{2}+\beta_{y}^{2}+\frac{i\omega }{v}\right)\right]}^{-1}\right\}}^{-1}$$
(11)$$C_{0e}=\frac{1}{P_{0}}\left\{1-\frac{4i\omega \left(\gamma -1\right)}{v^{\prime }a_{}^{2}h_{}^{2}}{\sum_{x=0}^{\infty }\sum_{y=0}^{\infty }\left[\alpha_{x}^{2}\beta_{y}^{2}\left(\alpha_{x}^{2}+\beta_{y}^{2}+\frac{i\omega \gamma }{v^{\prime }}\right)\right]}^{-1}\right\}$$
(12)$$v=\mu /\rho_{0}$$
(13)$$v^{\prime }=\kappa /\rho_{0}/C_{v}$$

According to the constructed theoretical model for the sound absorption coefficient of parallel–connection square Helmholtz resonators in the Equations (2)–(13), the theoretical sound absorption coefficients of the investigated metamaterial cells can be obtained, as shown in Figure 11, which are compared with the simulation data and the experimental data. It can be found that the difference between the theoretical data and the experimental data is larger than that between the simulation data and the experimental data, which further proves that effectiveness of the optimization method through the finite element simulation model. The major reason for low prediction accuracy in the theoretical model is that there are many approximations and omissions in the modeling process, which results in large fluctuations in the distribution of sound absorption coefficients within the effective frequency range and deviation of the absorption peak frequencies, as exhibited in Figure 11. For the three given conditions, the average absolute deviations of the simulation data and those of the theoretical data in the desired frequency ranges of [500 Hz, 800 Hz], [600 Hz, 900 Hz] and [700 Hz, 1000 Hz] are calculated by the Equations (14) and (15), respectively, and summarized in Table 5 by taking the experimental data as the standard, which can quantitatively prove that accuracy of the finite element simulation is better than that of the theoretical model.
(14)$$\begin{array}{cc} Dev1=average\left(\left|\alpha_{simulation}\left(f\right)-\alpha_{\exp erimental}\left(f\right)\right|\right) & f\in \left[f_{\min},f_{\max}\right] \end{array}$$
(15)$$\begin{array}{cc} Dev2=average\left(\left|\alpha_{theoretical}\left(f\right)-\alpha_{\exp erimental}\left(f\right)\right|\right) & f\in \left[f_{\min},f_{\max}\right] \end{array}$$

## 7. Conclusions

The studied metamaterial cell of parallel–connection square Helmholtz resonators is developed and optimized to obtain the low frequency noise control for the given conditions in the factory. The major conclusions achieved in this research are as follows:(1)Sound absorption coefficients of the investigated metamaterial cells obtained by the three–dimensional finite element model are consistent with those achieved by standing wave tube measurement, which can prove feasibility of the proposed optimization method and usefulness of the developed acoustic metamaterial absorber.(2)Actual sound absorption performance of the obtained optimal metamaterial cells can meet the requirements of the 3 given conditions in the experimental validation, which can prove the effectiveness of the initial values of parameters obtained by the two–dimensional equivalent simulation model and the accuracy of the optimal parameters achieved through the particle swarm optimization algorithm.(3)The average actual sound absorption coefficients of the three investigated metamaterial cells are 0.9271 in the [700 Hz, 1000 Hz] with total size of 30 mm, 0.9157 in the [600 Hz, 900 Hz] with total size of 40 mm, and 0.9259 in the [500 Hz, 800 Hz] with total size of 50 mm, respectively. Broadband sound absorption performance is obtained by the parameter optimization, which will be propitious for promoting the actual application of the proposed parallel–connection square Helmholtz resonators to reduce the low frequency noise generated by the large equipment in the factory.

## Figures and Tables

**Figure 1 materials-15-03417-f001:**
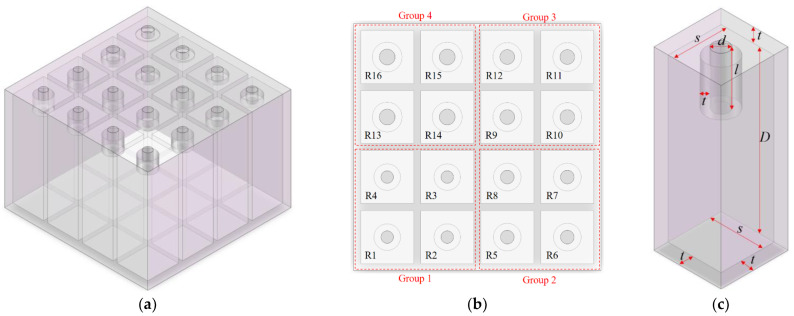
Schematic diagram of the metamaterial cell of the parallel–connection square Helmholtz resonators: (**a**) perspective three–dimensional model; (**b**) divination into groups and numbered resonators; (**c**) single resonator with marked parameters.

**Figure 2 materials-15-03417-f002:**
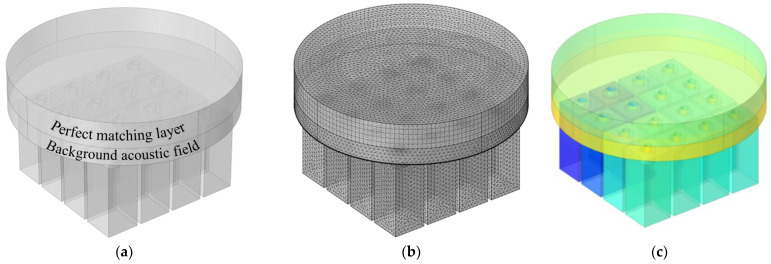
Three–dimensional finite element model for multi–groups parallel square Helmholtz resonators: (**a**) geometric model; (**b**) finite element mesh model; (**c**) sound pressure distribution.

**Figure 3 materials-15-03417-f003:**
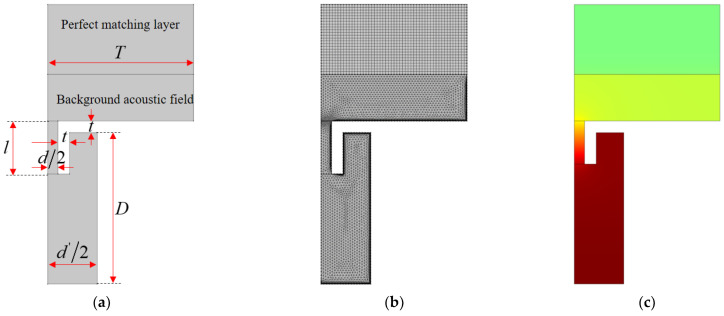
Two–dimensional equivalent simulation model for the single resonator in parallel–connection square Helmholtz resonators: (**a**) geometric model; (**b**) finite element mesh model; (**c**) sound pressure distribution.

**Figure 4 materials-15-03417-f004:**
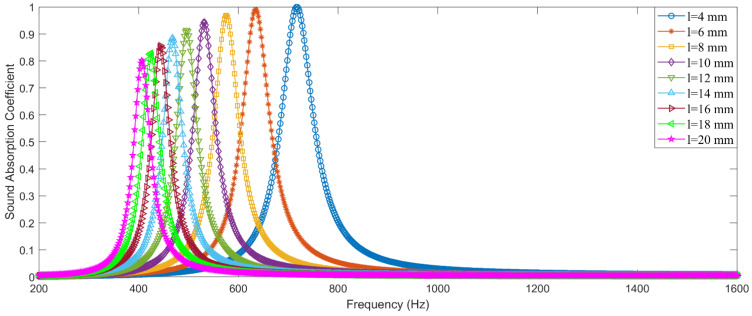
Distributions of theoretical sound absorption coefficients of single resonator along with increase in length of the aperture obtained by two–dimensional equivalent simulation model.

**Figure 5 materials-15-03417-f005:**
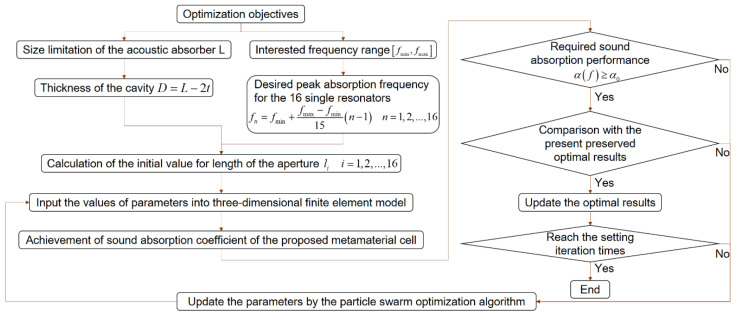
Flow chart of optimization process for the proposed metamaterial cell with given objectives.

**Figure 6 materials-15-03417-f006:**
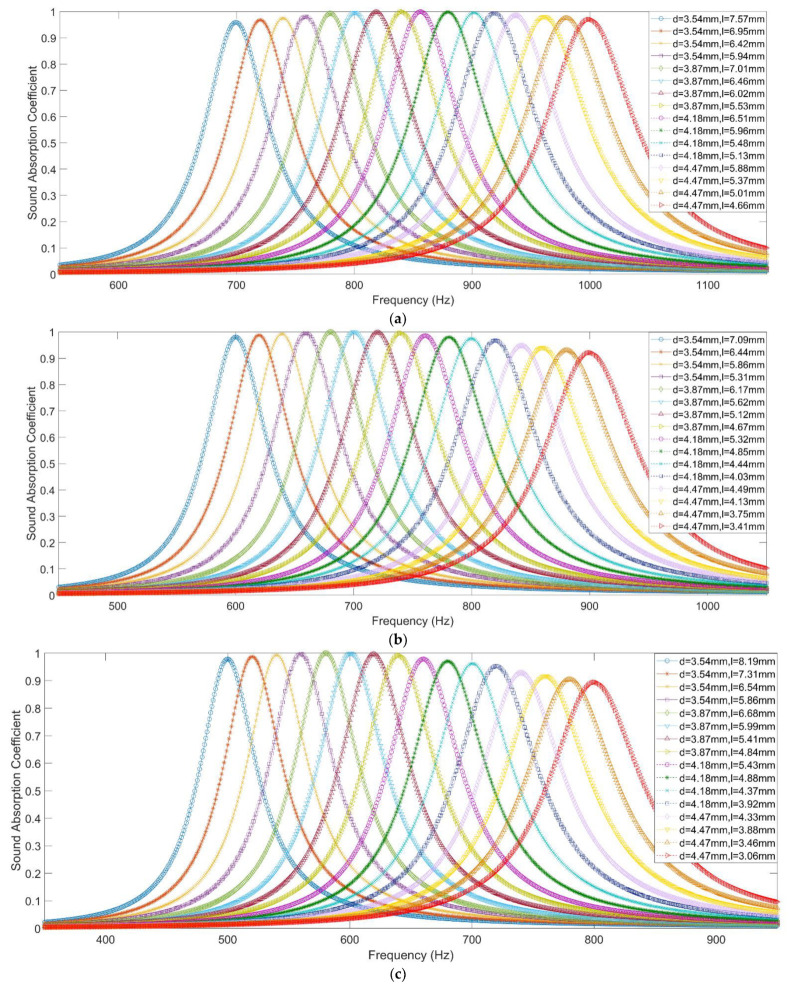
Distributions of sound absorption coefficients of each single resonator for the three conditions with initial values of parameters: (**a**) Condition–1; (**b**) Condition–2; (**c**) Condition–3.

**Figure 7 materials-15-03417-f007:**
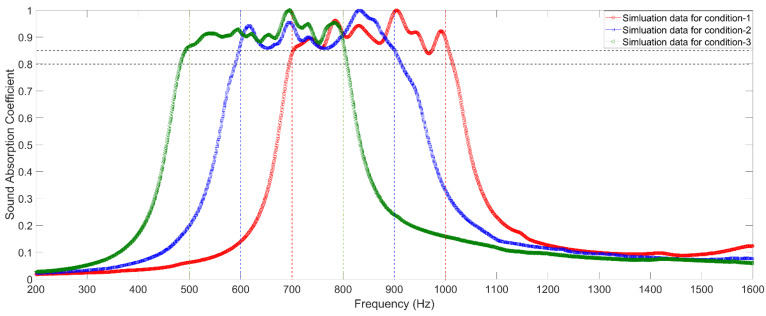
Distributions of theoretical sound absorption coefficients of the 3 optimized metamaterial cells obtained by the three–dimensional finite element model with the optimal parameters.

**Figure 8 materials-15-03417-f008:**
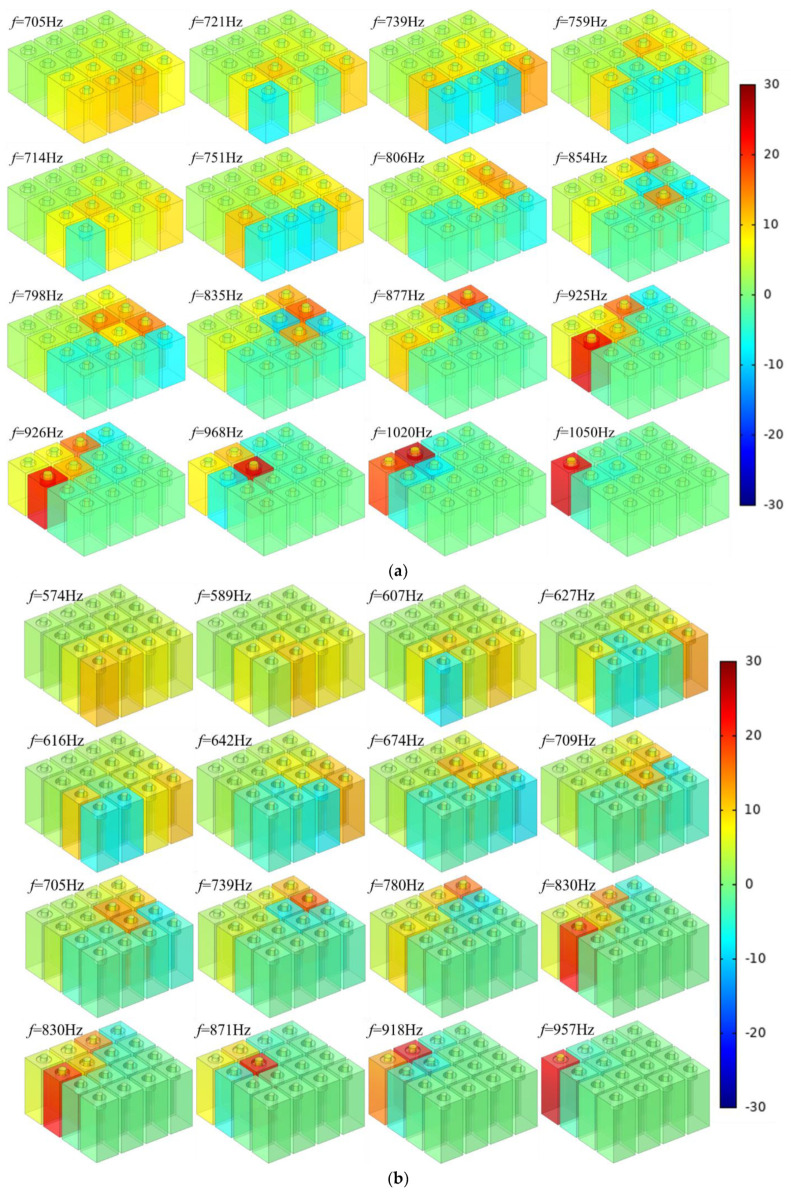
Distributions of the sound pressure at each resonance frequency for the three optimized metamaterial cells: (**a**) Condition–1; (**b**) Condition–2; (**c**) Condition–3.

**Figure 9 materials-15-03417-f009:**
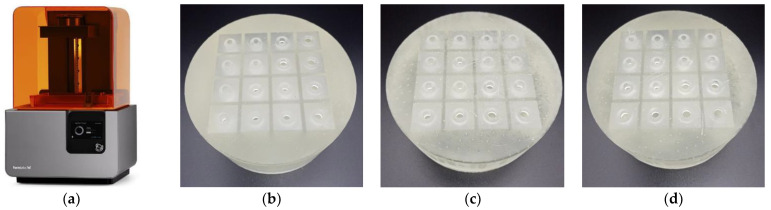
Preparation of experimental samples: (**a**) the utilized 3D printer; (**b**) sample for the condition–1; (**c**) sample for the condition–2; (**d**) sample for the condition–3.

**Figure 10 materials-15-03417-f010:**
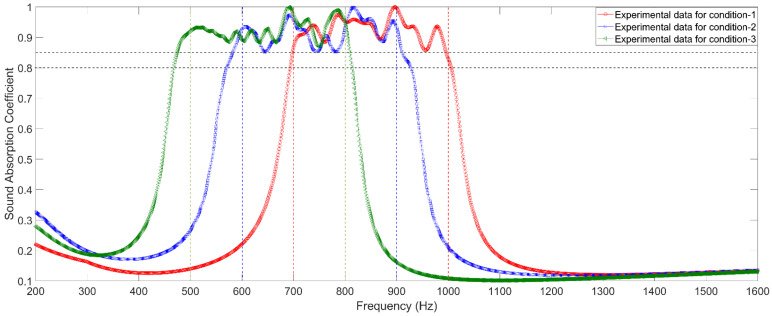
Distributions of actual sound absorption coefficients of the three optimized metamaterial cells obtained by the standing wave tube measurement.

**Figure 11 materials-15-03417-f011:**
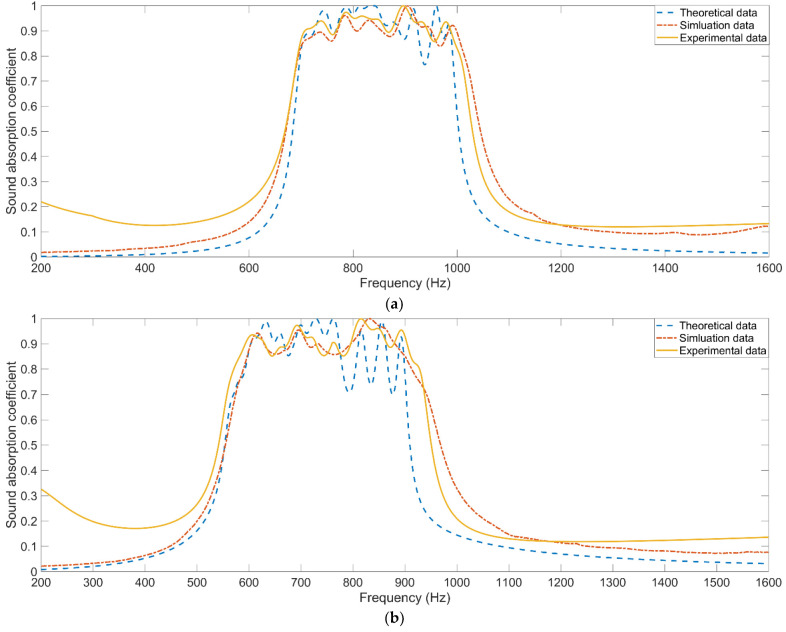
Comparisons of sound absorption coefficients of the 3 optimized metamaterial cells in theory, in simulation and in actual: (**a**) Condition–1; (**b**) Condition–2; (**c**) Condition–3.

**Table 1 materials-15-03417-t001:** Parameters of the metamaterial cell of the parallel–connection square Helmholtz resonators.

Number of Group	Number of Resonator	Parameters			
		Diameter of Hole	Length of Aperture	Side Length of Cavity	Thickness of Cavity
1	R1	3.54 mm	$l_{1}$	15 mm	*D*
	R2		$l_{2}$		
	R3		$l_{3}$		
	R4		$l_{4}$		
2	R5	3.87 mm	$l_{5}$		
	R6		$l_{6}$		
	R7		$l_{7}$		
	R8		$l_{8}$		
3	R9	4.18 mm	$l_{9}$		
	R10		$l_{10}$		
	R11		$l_{11}$		
	R12		$l_{12}$		
4	R13	4.47 mm	$l_{13}$		
	R14		$l_{14}$		
	R15		$l_{15}$		
	R16		$l_{16}$		

**Table 2 materials-15-03417-t002:** The three requirements for noise reduction in the given factories.

Optimization Objectives	Condition–1	Condition–2	Condition–3
Size limitation of the acoustic absorber	30 mm	40 mm	50 mm
Interested frequency range	700–1000 Hz	600–900 Hz	500–800 Hz
Required sound absorption performance	≥0.80	≥0.85	≥0.85
Average sound absorption coefficient	maximum	maximum	maximum

**Table 3 materials-15-03417-t003:** Initial values of parameters for the proposed metamaterial cell.

Condition–1			Condition–2			Condition–3		
Parameters		Absorption Peak	Parameters		Absorption Peak	Parameters		Absorption Peak
$l_{1}$	7.57	700 Hz	$l_{1}$	7.09	600 Hz	$l_{1}$	8.19	500 Hz
$l_{2}$	6.95	720 Hz	$l_{2}$	6.44	620 Hz	$l_{2}$	7.31	520 Hz
$l_{3}$	6.42	740 Hz	$l_{3}$	5.86	640 Hz	$l_{3}$	6.54	540 Hz
$l_{4}$	5.94	760 Hz	$l_{4}$	5.31	660 Hz	$l_{4}$	5.86	560 Hz
$l_{5}$	7.01	780 Hz	$l_{5}$	6.17	680 Hz	$l_{5}$	6.68	580 Hz
$l_{6}$	6.46	800 Hz	$l_{6}$	5.62	700 Hz	$l_{6}$	5.99	600 Hz
$l_{7}$	6.02	820 Hz	$l_{7}$	5.12	720 Hz	$l_{7}$	5.41	620 Hz
$l_{8}$	5.53	840 Hz	$l_{8}$	4.67	740 Hz	$l_{8}$	4.84	640 Hz
$l_{9}$	6.51	860 Hz	$l_{9}$	5.32	760 Hz	$l_{9}$	5.43	660 Hz
$l_{10}$	5.96	880 Hz	$l_{10}$	4.85	780 Hz	$l_{10}$	4.88	680 Hz
$l_{11}$	5.48	900 Hz	$l_{11}$	4.44	800 Hz	$l_{11}$	4.37	700 Hz
$l_{12}$	5.13	920 Hz	$l_{12}$	4.03	820 Hz	$l_{12}$	3.92	720 Hz
$l_{13}$	5.88	940 Hz	$l_{13}$	4.49	840 Hz	$l_{13}$	4.33	740 Hz
$l_{14}$	5.37	960 Hz	$l_{14}$	4.13	860 Hz	$l_{14}$	3.88	760 Hz
$l_{15}$	5.01	980 Hz	$l_{15}$	3.75	880 Hz	$l_{15}$	3.46	780 Hz
$l_{16}$	4.66	1000 Hz	$l_{16}$	3.41	900 Hz	$l_{16}$	3.06	800 Hz

**Table 4 materials-15-03417-t004:** Optimal results of parameters for the proposed metamaterial cell.

Condition–1			Condition–2			Condition–3		
Parameters		Absorption Peak	Parameters		Absorption Peak	Parameters		Absorption Peak
$l_{1}$	7.41	705 Hz	$l_{1}$	8.08	574 Hz	$l_{1}$	8.92	485 Hz
$l_{2}$	6.92	721 Hz	$l_{2}$	7.47	589 Hz	$l_{2}$	8.41	495 Hz
$l_{3}$	6.43	739 Hz	$l_{3}$	6.86	607 Hz	$l_{3}$	7.93	506 Hz
$l_{4}$	5.92	759 Hz	$l_{4}$	6.23	627 Hz	$l_{4}$	7.44	517 Hz
$l_{5}$	9.11	714 Hz	$l_{5}$	8.36	616 Hz	$l_{5}$	8.81	529 Hz
$l_{6}$	7.84	751 Hz	$l_{6}$	7.37	642 Hz	$l_{6}$	8.02	546 Hz
$l_{7}$	6.32	806 Hz	$l_{7}$	6.36	674 Hz	$l_{7}$	7.21	566 Hz
$l_{8}$	5.23	854 Hz	$l_{8}$	5.39	709 Hz	$l_{8}$	6.43	587 Hz
$l_{9}$	8.13	798 Hz	$l_{9}$	6.84	705 Hz	$l_{9}$	6.92	614 Hz
$l_{10}$	7.04	835 Hz	$l_{10}$	5.88	739 Hz	$l_{10}$	5.91	644 Hz
$l_{11}$	6.02	877 Hz	$l_{11}$	4.86	780 Hz	$l_{11}$	4.93	678 Hz
$l_{12}$	5.01	925 Hz	$l_{12}$	3.85	830 Hz	$l_{12}$	3.94	719 Hz
$l_{13}$	6.12	926 Hz	$l_{13}$	4.74	830 Hz	$l_{13}$	4.82	720 Hz
$l_{14}$	5.24	968 Hz	$l_{14}$	3.92	871 Hz	$l_{14}$	3.84	762 Hz
$l_{15}$	4.31	1020 Hz	$l_{15}$	3.11	918 Hz	$l_{15}$	2.81	814 Hz
$l_{16}$	3.83	1050 Hz	$l_{16}$	2.52	957 Hz	$l_{16}$	2.53	829 Hz

**Table 5 materials-15-03417-t005:** The average absolute deviations of simulation data and theoretical data by comparison with experimental data.

Condition Serial	Interested Frequency Range	Average Absolute Deviations	
		For simulation Data	For Theoretical Data
Condition–1	700–1000 Hz	0.0571	0.1086
Condition–2	600–900 Hz	0.0685	0.1305
Condition–3	500–800 Hz	0.0553	0.1147

## Data Availability

The data that support the findings of this study are available from the corresponding author upon reasonable request.

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
