# Peer review of "Development and Optimization of Broadband Acoustic Metamaterial Absorber Based on Parallel–Connection Square Helmholtz Resonators"

_materials, 2022, doi:10.3390/ma15103417_

Round 1

Reviewer 1 Report

This paper proposes a metamaterial composed by s multi-series parallel square Helmholtz resonators. The paper can be published after minor revisions. Below are some specific comments.

  • Similar attempts have been made using MPP and other sound absorbing systems. Therefore, the authors should try to state the contribution and the merit of this work more clearly.
  • Line 126: Definition of perforation ratio in this case should be clearly stated. Also, the evidence of its range 0.5 -0.9% should be explained.
  • Details of FEM simulation should be more clearly stated, e.g., mesh type, size, etc.
  • Is the FE model validated? or evaluated with analytical solution?
  • Is there any interaction between resonators in some cases?
  • References are rather limited and biased. It is recommendable for the authors to include more similar works.

Author Response

Thank you very much for your kind review to our manuscript and positive assessment to our research. We have revised the manuscript carefully according to your and other reviewers’ comment. The responses to your comments are as follows.

  1. Similar attempts have been made using MPP and other sound absorbing systems. Therefore, the authors should try to state the contribution and the merit of this work more clearly.

Response: Thank you very much for your comment. As mentioned in the introduction part, there are three difficulties in development and optimization of the parallel square Helmholtz resonators. First of all, the sound absorption coefficient of the parallel square Helmholtz resonators is difficult to be calculated accurately, because the theoretical sound ab-sorption model is imprecise for some simplifications, approximations, hypothesis and abbreviations. Although the finite element simulation can achieve high prediction accuracy, it takes long time. Secondly, there are many structural parameters need to be optimized for the parallel square Helmholtz resonators, and increase in the number of parameters not only raises the difficulty in design and optimization, but also extends the calculation or simulation time. Thirdly, an effective optimization process not only requires a suitable utilized algorithm, but also be determined by the selected initial value, the given constraint conditions and desired optimization objectives.

Therefore, the parallel-connection square Helmholtz resonators is developed and investigated in this study, which aims to overcome the mentioned 3 difficulties. The research achievements obtained in this research can prove feasibility of the proposed optimization method and usefulness of the developed acoustic metamaterial absorber.

The contribution and the merit of this work are stated in the conclusion part. The modifications are added in the revised manuscript and highlighted in yellow.

  1. Line 126: Definition of perforation ratio in this case should be clearly stated. Also, the evidence of its range 0.5 -0.9% should be explained.

Response: Thank you very much for your suggestion. The perforation rate r is defined as the ratio of total area of the holes with the same diameter to whole area of the metamaterial cell, as shown in the Equation (1) in the manuscript, which is similar to definition of perforation rate for the microperforated panel absorber.

If the perforation rate is larger than 0.9%, the resonance frequency will exceed the low-middle frequency range, which cannot show advantages of Helmholtz resonator in the low frequency region. Meanwhile, when the perforation rate is smaller than 0.5%, the thermal viscous effect and thermal conduction effect will be weakened, which leads to the decrease of peak sound absorption coefficient, Therefore, the perforation rate should be in the reasonable range. These modifications are added in the revised manuscript and highlighted in yellow.

  1. Details of FEM simulation should be more clearly stated, e.g., mesh type, size, etc.

Response: Thank you very much for your suggestion. Details of finite element simulation model is stated in the part 3 “Construction of the Finite Element Model”, including the mesh type, size, layer number, stretching factor, and so on.

The parameters used in the three-dimensional finite element model are as follows: size of the largest unit, 2 mm; size of the smallest unit, 0.02 mm; maximum growth rate of neighboring unit, 1.3; curvature factor, 0.2; resolution ratio of the narrow area, 1; mesh type, free tetrahedron mesh; layer number of the boundary area, 8; stretching factor of the boundary layer, 1.2; regulatory factor of the thickness of the boundary layer, 1.

These modifications are added in the revised manuscript and highlighted in yellow.

  1. Is the FE model validated? or evaluated with analytical solution?

Response: Thank you very much for your comment. Judging from comparisons of sound absorption coefficients of the 3 optimized metamaterial cells in theory, in simulation and in actual in the Figure 11, it can be found that the difference between the theoretical data with the experimental data is larger than that between the simulation data with the experimental data, and the simulation data exhibited good consistency with the experimental data. Thus, effectiveness of the finite element simulation model can be validated and proved.

  1. Is there any interaction between resonators in some cases?

Response: Thank you very much for your comment. There exist interaction between different resonators, especially for the resonators in one group. Judging from calculation formula for perforation rate r in the Equation (1), it can be found that the resonators in one group have the same diameter of the hole and they share the same perforation rate. Meanwhile, diameters of the hole for 4 groups of resonators in the metamaterial cell can be calculated as 3.54 mm, 3.87 mm, 4.18 mm and 4.47 mm respectively, and it can be found that differences among the resonators in different groups are not so big. Thus, there also exist interaction among all the resonators. The interaction will be stronger when parameters of the resonators are close to each other, and it will be weaker when parameters of the resonators are far away from each other.

  1. References are rather limited and biased. It is recommendable for the authors to include more similar works.

Response: Thank you very much for your comment. Some related similar articles are added in the reference lists.

Reviewer 2 Report

The submitted manuscript deals with the design of a metasurface consisting of cells formed by detuned Helmholtz resonators. The parameters of the Helmholtz resonators are determined by means of an optimization approach in order to provide high absorption coefficient in a wide frequency range. The results are presented in form of 3D numerical simulations as well as experimental data. The subject of the manuscript is quite topical and it has attracted attention in recent years, that is why I recommend its publication in Materials. However, there are issues which should be addressed first.

  1. I strongly recommend not to use the term “series” in the connection with the arrangement of the Helmholtz resonators. In acoustics (especially in muffler design), this term is used when the resonators are attached one to another. In this case, all the resonators are arranged in parallel, and the term “series” is only confusing.
  2. The authors conduct numerical simulations, which are a substantial part of the paper. However, they do not provide any detailed information about the used mathematical model, which is crucial in this context. How are the losses of acoustic energy introduced? Do the authors employ linearized Navier-Stokes equations, or Helmholtz equation and some kind of an effective equivalent media?
  3. Figure 8 – it would probably be more clear to plot the acoustic pressure amplitude to see in which resonators the resonance takes place.
  4. The article definitely should be read by a native English speaker, there is a plenty of grammar errors which make the article hard to read and some sentences can be hardly understood.

Author Response

Thank you very much for your kind review to our manuscript and positive assessment to our research. We have revised the manuscript carefully according to your and other reviewers’ comment. The responses to your comments are as follows.

  1. I strongly recommend not to use the term “series” in the connection with the arrangement of the Helmholtz resonators. In acoustics (especially in muffler design), this term is used when the resonators are attached one to another. In this case, all the resonators are arranged in parallel, and the term “series” is only confusing.

Response: Thank you very much for your suggestion. The description of “Multi-series Parallel Square Helmholtz Resonators” has been replaced by “Parallel-Connection Square Helmholtz Resonators”, both in the title and the text, and these modifications are highlighted in yellow.

  1. The authors conduct numerical simulations, which are a substantial part of the paper. However, they do not provide any detailed information about the used mathematical model, which is crucial in this context. How are the losses of acoustic energy introduced? Do the authors employ linearized Navier-Stokes equations, or Helmholtz equation and some kind of an effective equivalent media?

Response: Thank you very much for your kind comment. Limited by length of the original manuscript, detailed information about the used mathematical model for the finite element simulation is not given. The finite element simulation model is constructed in the COMSOL Multiphysics software and thermal viscous acoustic model in the acoustic module is utilized. When the frequency of incident sound wave is consistent with the resonance frequency of resonant cavity, there will be frequent expansion and compression for air in the cavity, which can results in reciprocating motion with high speed for air column in the aperture. Thermal viscosity effect between air column in the aperture and the boundary and thermal conductivity effect in the wall can lead to the losses of acoustic energy, which achieves the sound absorption effect.

These modifications are added in the revised manuscript and highlighted in yellow.

  1. Figure 8 – it would probably be more clear to plot the acoustic pressure amplitude to see in which resonators the resonance takes place.

Response: Thank you very much for your suggestion. It could be observed from distribution of the sound pressure in the Figure 8 that color of the resonant cavity at its resonance frequency is obviously different from the other cavities, so it can exhibit the resonance takes place in which resonators. Meanwhile, distributions of the sound pressure is the normally method to judge the resonant cavity in many references, so we use this method in this research. In future, we will attempt to use the acoustic pressure amplitude to judge the resonant cavity.

  1. The article definitely should be read by a native English speaker, there is a plenty of grammar errors which make the article hard to read and some sentences can be hardly understood.

Response: Thank you very much for your suggestion. The manuscript is revised by a native English speakers to eliminate the grammar or spelling mistakes and to make the paper more readable

Reviewer 3 Report

The acoustic metamaterial absorber of multi-series parallel square Helmholtz resonators is proposed in this study and its sound absorption coefficients are optimization to reduce the noise for the given conditions in the factory. I think paper is interesting, my minor changes are as follows:

  1. Authors should clearly describe origin of the absorption.
  2. Authors are missing some articles in the field Analytic solution to field distribution in one-dimensional inhomogeneous media, etc.
  3. The simulation results have been performed finite element method. Authors should justify the choice of this methodology in the manuscript including explanation why derivation of the analytical expressions is not possible in the context of the solved problem.
  4. Authors should comment additionally on the “theoretical data” concept used throughout the manuscript. Is it used aiming to approbate the achieved simulation results?

Author Response

Thank you very much for your kind review to our manuscript and positive assessment to our research. We have revised the manuscript carefully according to your and other reviewers’ comment. The responses to your comments are as follows.

  1. Authors should clearly describe origin of the absorption.

Response: Thank you very much for your suggestion. Sound absorption mechanism of the Helmholtz resonator is thermal viscosity effect and thermal conductivity effect. When the frequency of incident sound wave is consistent with the resonance frequency of resonant cavity, there will be frequent expansion and compression for air in the cavity, which can results in the reciprocating motion with high speed for air column in the aperture. Thermal viscosity effect between air column in the aperture and the boundary and thermal conductivity effect in the wall can lead to the losses of acoustic energy, which achieves the sound absorption effect.

These modifications are added in the revised manuscript and highlighted in yellow.

  1. Authors are missing some articles in the field Analytic solution to field distribution in one-dimensional inhomogeneous media, etc.

Response: Thank you very much for your suggestion. Some references are adjusted and added, which aim to make the manuscript more reasonable.

  1. The simulation results have been performed finite element method. Authors should justify the choice of this methodology in the manuscript including explanation why derivation of the analytical expressions is not possible in the context of the solved problem.

Response: Thank you very much for your comment. In this study, we use the finite element simulation instead of derivation of the analytical expressions to predict sound absorption performance of the proposed metamaterial cell. The major reason has been stated in the paragraph 6 in the “6. Analysis and Discussion” part. The major reason for low prediction accuracy in the theoretical model is that there are many approximations and neglects in the modeling process, which results in large fluctuations in the distribution of sound absorption coefficients within the effective frequency range and deviation of the absorption peak frequencies, as exhibited in the Figure 11.

  1. Authors should comment additionally on the “theoretical data” concept used throughout the manuscript. Is it used aiming to approbate the achieved simulation results?

Response: Thank you very much for your comment. It has been proved by many present researches that the prediction accuracy of theoretical model is low, because the theoretical sound absorption model is imprecise for some simplifications, approximations, hypothesis and abbreviations. Therefore, we only mentioned the theoretical data in the “6. Analysis and Discussion” part. The theoretical data just is used to explain why we use the finite element simulation to predict the sound absorption coefficient of metamaterial cell, because prediction accuracy of the theoretical model is obviously worse than that of the finite element simulation.